# A Comparative Review of Pregnancy and Cancer and Their Association with Endoplasmic Reticulum Aminopeptidase 1 and 2

**DOI:** 10.3390/ijms24043454

**Published:** 2023-02-09

**Authors:** Brian Hur, Veronica Wong, Eun D. Lee

**Affiliations:** 1Department of Microbiology and Immunology, School of Medicine, Virginia Commonwealth University, Richmond, VA 23298, USA; 2Department of Biology, Virginia Commonwealth University, Richmond, VA 23298, USA

**Keywords:** pregnancy, cancer, ERAP, angiogenesis, migration, immune cells, peptide process

## Abstract

The fundamental basis of pregnancy and cancer is to determine the fate of the survival or the death of humanity. However, the development of fetuses and tumors share many similarities and differences, making them two sides of the same coin. This review presents an overview of the similarities and differences between pregnancy and cancer. In addition, we will also discuss the critical roles that Endoplasmic Reticulum Aminopeptidase (ERAP) 1 and 2 may play in the immune system, cell migration, and angiogenesis, all of which are essential for fetal and tumor development. Even though the comprehensive understanding of ERAP2 lags that of ERAP1 due to the lack of an animal model, recent studies have shown that both enzymes are associated with an increased risk of several diseases, including pregnancy disorder pre-eclampsia (PE), recurrent miscarriages, and cancer. The exact mechanisms in both pregnancy and cancer need to be elucidated. Therefore, a deeper understanding of ERAP’s role in diseases can make it a potential therapeutic target for pregnancy complications and cancer and offer greater insight into its impact on the immune system.

## 1. Introduction

The human immune system is a complex collection of organs, cells, and proteins that protect the body from invading foreign or unhealthy self-entities. It uses various methods to achieve protection, including using peptides and the human leukocyte antigen (HLA) system to detect and eliminate components that are harmful to the body [1]. The HLA molecules regulate the immune system through bound peptides that are presented on the cell surface, thus enabling immune cells to survey the peptides to differentiate between normal and unhealthy cells. While the immune system relies on the HLA system to recover from illnesses, the same system can induce immune tolerance, causing an increase in the likelihood of developing disease [2] or increasing fetal survival during pregnancy [3]. Between the two scenarios, pregnancy and cancer share many similar modes of tolerance, which can be achieved through irregular peptides and abnormal HLA presentation.

Endoplasmic Reticulum Aminopeptidase 1 and 2 (ERAP1 and ERAP2, respectively) are enzymes in the endoplasmic reticulum responsible for peptide processing and presentation. They are critical in regulating and determining the peptide repertoire [4]. In addition to their involvement in the immune system, ERAP1, and ERAP2 also play a role in cell migration and angiogenesis, which are necessary for both pregnancy and cancer [5]. A comparison of the similarities and differences between pregnancy and cancer and the roles ERAP1 and ERAP2 play in their progression is analyzed further. 

The development of a fetus during the first trimester of pregnancy is very similar to the growth of tumor cells. Both fetuses and tumors have a different genetic makeup from the self and should be recognized by the host’s immune system [6]. However, both use mechanisms to induce a tolerogenic immune environment for survival. The fetus uses many similar pathways to proliferate, invade, and control the immune system that the tumor cells use [7]. Comparing the two conditions can allow us to gain insights into developing new treatments for pregnancy complications and preventing tumor growth. Such pathways found in one scenario can be tested in the other, which may lead to new immunotherapies to treat the difficulties experienced during pregnancy and cancer. 

The differences between pregnancy and cancer can be used to determine the threshold of proliferation, invasion, and immune modulation allowed before it is characterized as detrimental to the body. This review explores the effects of peptide processing and presentation on the immune responses through the presence or absence of ERAP1 and ERAP2. ERAP1 and ERAP2’s structure, location, and function implicate their potential impact on the irregular peptides, and HLAs found in trophoblast and tumor cells. These implications can modulate immune responses that can either promote the tolerance for survival or the activation for elimination.

## 2. Similarities

Pregnancy and cancer are very similar as they begin with single cells in the body that use similar methods to proliferate, invade, and manipulate the immune system to their advantage. Both use common pathways, proteins, and hormones to secure resources and grow.

### 2.1. Proliferation

Proliferation is the growth and division of cells to produce a greater mass of tissues. Both trophoblast and tumor cells proliferate rapidly to increase their numbers. Trophoblast cells are the outermost cell layer of the blastocyst that invade, implant and develop in the placenta, which is responsible for maternal and fetal oxygen and nutrient exchange. Telomerase, an enzyme that increases proliferation due to its maintenance of the telomere length, activity is at its highest during the first trimester when most cell division occurs, and it is increased for up until 20 weeks during gestation [8]. Eighty-five percent of cancers show increased telomerase activity, suggesting that telomerase is vital for increased tumor proliferation [3,9]. Survivin is another protein used to increase proliferation; a low expression causes chromosome misalignment and a loss of centrosome integrity, which impairs cell division [10]. Survivin is vital for cell division, which supports the findings of upregulation in trophoblasts and overexpression in cancers [3], as well as an association between the resistance to chemotherapy and the recurrence of tumors [8]. 

Growth factor pathways are also vital for proliferative activity. The insulin-like growth factor (IGF), epidermal growth factor (EGF), hepatocyte growth factor (HGF), and vascular endothelial growth factor (VEGF) are some major growth factor pathways used in trophoblast and cancer cells. The IGF1R pathway increases mitosis, decreases apoptosis, and protects the tumor cells from chemotherapy and radiation [3]. Similarly, the IGF pathway regulates placental growth and amino acid transport and is vital for the fetus’ survival [11]. The production of EGF is increased in trophoblast and carcinoma cells. In cancers, the HGF pathway is associated with growth and metastasis, while in trophoblasts, it is associated with proliferation and survival. The VEGF pathway is an important pathway that is used often in both the proliferative and invasive stages of pregnancy and cancer. As it is expressed in cytotrophoblasts, and fetal and maternal macrophages, VEGF regulates cell migration, survival, angiogenesis, and proliferation. It is also associated with angiogenesis and proliferation in tumor cells [12]. The mitogen-activated protein kinase pathway (MAPK) is an additional pathway that is used in proliferation. It is vital for cancer cell proliferation, growth, and resistance to drug treatments [13], while supporting trophoblast expansion [14]. 

The mammalian target of the rapamycin (MTOR) pathway regulates trophoblast proliferation and is activated in many malignancies [3]. Suppressors of mothers against decapentaplegic (SMAD) proteins regulate the proliferative phenotypes in trophoblasts and increase proliferation in cancer cells through the transforming growth factor (TGF) pathway [15]. Female estrogen is another method that is used for proliferation, for instance, estrogen has many effects on proliferation throughout pregnancy [16], and specific estrogen receptors are markers of breast cancer in females [17]. Endoreduplication, another method that tumors and trophoblasts use, occurs when chromosomes replicate, but the separation of chromatids does not happen, resulting in a polyploid cell. It leads to resistance to DNA-damaging agents such as chemotherapy or radiation in tumors and trophoblasts, which allow proliferation to go unhindered [3]. Proliferation is a complex process with many parts; there may be more activities that pregnancy and cancer share.

### 2.2. Cellular Invasion and Angiogenesis for Blood Supply

Invasion and blood supply are vital for the growth of trophoblasts and tumor cells, especially the metastasis of the tumors. Proliferation is an intensive process requiring trophoblast and tumor cells to invade the tissue and establish a blood supply, which both types of cells do by encouraging angiogenesis and the formation of new blood vessels. For example, both cells perform the epithelial-to-mesenchymal transition (EMT), which allows cell contact, which is vital for invasion [18]. Trophoblast and tumor cells will also emulate epithelial cells to achieve angiogenesis. Migration-inducing gene 7 (MIG-7) enables trophoblasts and tumor cells to invade, as it is present in most epithelial-emulating tumor cells and invasive extravillous trophoblasts (EVTs) [19]. Matrix glycoprotein-binding galectin 3 is overexpressed in EVTs and helps aggressive melanomas to develop an endothelial phenotype [20]. 

Invasion requires many pathways in order for it to occur successfully. For example, the wingless T cell (WNT) pathway changes EVTs from a proliferative phenotype to an invasive phenotype [3] and causes endometrial changes in implantation, such as endometrial gland formation [21]. WNT also increases motility, proliferation [3], and T-cell exclusion in some cancers [22]. Another pathway involved in invasion is the VEGF pathway, which increases angiogenesis, resulting in uncontrolled growth around the tumor [23] and promoting implantation and vascular formation in the placenta [24]. The MTOR pathway is correlated with angiogenesis in trophoblast and malignancies [3]. The phosphatidylinositol 3-kinase/protein kinase B (P13K/AKT) pathway is necessary for blastocyst hatching, the preparatory stage for implantation [25], and tumor development [26]. The janus kinase signal transducer activator of the transcription (JAK-STAT) pathway and focal adhesion kinase (FAK) activity enhances the invasiveness of tumors and trophoblast cells [15,27].

Trophoblast and tumor cells both share proteins that are necessary for invasion. Heat shock protein 27 (HSP27) is elevated in migratory EVTs and tumor cells, preventing the remodeling of the actin cytoskeleton, increasing the metastatic tumor potential, and inhibiting apoptosis [3]. Matrix metalloproteinases (MMPs) play a role in neovascularization, tumor metastasis, and uterine and vascular remodeling in trophoblasts [28,29]. Ras homolog family member A (RhoA) participates in angiogenesis and proliferation in cancer [30], and the RhoA-rho-associated protein kinase (Rock) signaling cascade is involved in trophoblast migration [15]. In tumors, Galectin-1 is associated with tumor formation, metastasis, angiogenesis, and apoptosis [31], while in trophoblasts, it regulates trophoblast invasion, maternal immune regulation, and angiogenesis [32]. 

Trophoblast and tumor cells also share invasive factors such as protease-activated receptor-1 (PAR1), which is associated with proliferation and invasion in EVTs and induces endothelial cell activation and communication, enabling tumors invasive capabilities [33]. Similarly, angiopoietin is vital for spinal arterial remodeling during pregnancy and tumor growth [34]. In contrast, pigment epithelium-derived factor (PEDF) restricts the growth and expansion of the fetal and placental endothelium [35] and is antiangiogenic and antitumorigenic in some cancers [36]. 

Trophoblast and tumor cells require a variety of pathways to correctly complete angiogenesis and invasion, further highlighting how crucial these processes are to the cells. While the comparison between trophoblast invasion and cancer has been previously recognized by other researchers, exploring both the pathways and making connections for future applications of cancer treatments that target the attack mechanisms can lead to more successful results due to a better understanding of the pathophysiology of both concepts [37,38,39]. 

### 2.3. Immune Tolerance

Fetuses and tumors have different genotypes compared to those of the host cells. To survive, they must escape the immune system by altering the function of the immune cells; for example, natural killer (NK) cells become more immunomodulatory during pregnancy and cancer [40]. Uterine NK cells do not perform cell-mediated cytotoxicity as peripheral NK cells do. In contrast, in a similar mechanism, NK cells in the tumor environment are suppressed by cytokines and are rendered dysfunctional [41]. Pregnancy and cancer promote proangiogenic, immature dendritic cells (DCs), enabling easier proliferation and invasion. While trophoblast angiogenesis is reduced in decreased DC maturation, tumor cells conversely secrete VEGF, TGF-β, and osteopontin to compensate for it. Tumor and uterine macrophages are immunosuppressive and secrete Th2 immunomodulatory cytokines [42]. Indoleamine (IDO) inhibits the proliferation of lymphocytes in the tumor and fetal microenvironments. Macrophage migration inhibitory factor (MIF) prevents abnormal trophoblast apoptosis [43] and increases tumor aggression [44].

The type and amount of human leukocyte antigens (HLAs) are also similar in trophoblast and tumor cells. Both lack common HLA class I molecules, such as HLA-A and HLA-B. Down-regulated or absent HLA class I’s can lead to cloaking from the immune system, which is why trophoblasts and tumor cells have limited or no expression of common HLA class I molecules. On the other hand, both have high levels of HLA-G, which is known to suppress NK and T cell-induced apoptosis, regulate cytokine production, and suppress DC maturation [3,45]. In addition, both trophoblasts and tumor cells secrete soluble HLA-G into the blood, impairing NK/DC crosstalk, promoting proinflammatory cytokine secretion within uterine mononuclear cells, and inducing apoptosis in CD8^+^ T cells [46].

Trophoblast and tumor cells also use several cytokines to regulate the immune system. They promote cytokines that increase the tolerogenicity of the microenvironment. Trophoblast and tumor cells prefer T_H_2-dominant cytokines (immunotolerant) over T_H_1-dominant cytokines (inflammatory) [47]. One example is CD200; it inhibits CD8^+^ T lymphocytes, shifts the cytokine balance in trophoblastic cells, and down-regulates cytokines in solid tumors such as carcinomas [48]. Growth-regulated alpha (Gro-A), metallocarboxypeptidase inhibitor (MCPI), and interleukin 8 (IL-8) recruit tolerogenic CD14+ monocytes to the fetal–maternal interface [49]. Gro-A is an oncogenic and angiogenic cytokine highly activated in many cancers [50]. MCPI increases the angiogenesis and recruitment of tumor macrophages. IL-8 is an immune and angiogenic factor in the tumor microenvironment that plays a role in chemotherapy resistance [51]. Some melanomas induce the secretion of C-C motif chemokine ligand 5 (CCL5), a chemokine that participates in the apoptosis of harmful maternal CD3+ cells in tumor-infiltrating lymphocytes, which causes the cells to undergo apoptosis as a method to avoid immune rejection [52]. CCL5 promotes monocyte migration into breast tumor sites, furthering its progression [53]. Ultimately, more research into the immune aspect can lead to successful immunotherapies with minimal side effects.

## 3. Differences

### 3.1. Alteration in Degrees of Regulation

The differences between pregnancy and cancer can be observed in the molecular pathways in both environments. With the IGF pathway, increases in proliferation occur in both pregnancy and cancer. However, in a pregnancy setting, IGF is regulated by IGF-binding proteins and proteases, while this mechanism is not present in cancer [3]. Another comparison is that the fetal form of the insulin receptor-a (IR-A), which increases proliferation and survival, is controlled in the fetus but not in cancer cells [54,55,56]. Tumors mainly express T_H_2 cytokines throughout, while trophoblasts express T_H_1 and T_H_2, mainly T_H_2, during gestation [2]. Differences in the TH1 and TH2 response regulation may affect the cells’ proliferative, invasive, and modulative capacities. Pregnancy is more tightly regulated than cancer, so pregnancy is often successful with little to no harm to the mother. In contrast, cancer is uncontrolled and can be deadly.

### 3.2. Microenvironment and Immune System

There are many notable differences in the regulatory T cells and other aspects of the uterine and tumor microenvironments. Regulatory T cells (Tregs) suppress the immune system to maintain homeostasis and self-tolerance to the fetus and the tumors. Even though they serve the same purpose, both uterine and tumor Tregs differ in their response to antigens, proliferation mechanisms, and the release of different cytokines. Uterine Tregs respond to maternal and fetal alloantigens, which include some non-self-antigens. However, tumor Tregs respond to self and neoantigens rather than non-self-antigens. Uterine Tregs proliferate locally because of fetal alloantigens, while tumor Tregs migrate towards the tumor. All uterine Tregs are immunosuppressive, while tumor Tregs can be immunosuppressive Foxp3^hi^ Tregs and non-suppressive Foxp3^lo^ Tregs [2]. The tumor’s microenvironment contains many extracellular supporting components, such as fibroblasts, pericytes, adipocytes, and extracellular matrix components, not found in the uterine environment. In addition, the tumor microenvironment uses myeloid cells, leukocytes that initiate immune responses, for angiogenesis and immunity [57]. On the other hand, the uterine microenvironment uses myeloid cells only for immunity purposes [58]. 

Many proteins, hormones, and cytokines play diverging roles in pregnancy and cancer. For instance, a carcinoembryonic antigen-related cell adhesion molecule 1 (CEACAM-1) inhibits NK cytolysis in trophoblasts while it increases angiogenesis and metastasis in cancers [3]. Osteopontin is vital for blastocyst proliferation and implantation [59]. Still, it is chemotactic for proinflammatory macrophages, T cells, and DCs in cancers, and thus, it is overexpressed in cancer but not in pregnancy [3]. Glycodelin has many functions and roles in cancer, such as proliferation, invasion, angiogenesis, and modulating immune cells (T cells, DCs, NK cells, B cells, and macrophages) [60]. Glycodelin-A is only used as a paracrine regulator in early pregnancy [61]. Despite their similarities, pregnancy, and cancer involve pathways, immune responses, proteins, hormones, and cytokines that play juxtaposing roles.

## 4. ERAP1 and ERAP2

### 4.1. ERAP Structure

Endoplasmic Reticulum Aminopeptidase 1, ERAP1, comprises 20 exons and spans over 47 kb. The gene-coded protein consists of four main domains. Of the four domains, domain II adopts a thermolysin-like catalytic domain that contains the protein’s active site with the consensus sequence GAMEN and zinc binding/gluzincin motifs [62]. 

ERAP1 protein function depends on the aminopeptidase’s conformational states, where it is presented in either open or closed forms. During the sealed condition, domain IV is oriented away from domain II, producing an active catalytic pocket with domains I, II, and IV, forming a large cavity. The difference between the two conformational states is attributed to the variation in the conformation of domain III [62]. 

Functionally, ERAP1 is an aminopeptidase involved in trimming long peptide molecules to the required lengths to present MHC Class I molecules. Induced by INF-y, ERAP1 has substrate preferences for hydrophobic residues or peptides containing aliphatic N-terminal amino acids [4,63]. ERAP1 can cleave all the peptide bonds except those involving proline. However, aminopeptidase shows a wide range of enzymatic efficiency depending on the N-terminal side chain of the substrate, showing a preferential cleavage of peptides that are longer than 9-mer. Peptide molecules that are shorter than 9-mers will not be processed by ERAP1, and the aminopeptidase will be inactive with shorter peptides [64]. As a result, it is thought that ERAP1 is the main trimming enzyme of the ER and is responsible for processing peptides of the required length. Additionally, RNAi studies have suggested that ERAP1 may be involved in forming about one-third of the peptide MHC Class I complexes [4,63]. 

ERAP2 is a 10-exon gene located on chromosome 5q15 between ERAP1 and the leucyl-cysteinyl aminopeptidase (LNPEP) gene. The gene is in the opposite orientation and likely shares the same regulatory elements as the surrounding genes [65]. The ERAP2 gene codes for zinc-metallopeptidases consisting of four globular domains. Of the four domains, catalytic activity occurs in domain II within the protein cavity, while substrate binding activity occurs in parts II and IV [64]. The catalytic site contains a lysine residue, participating in enzyme-product complex formation. It is adjacent to a large internal cavity to accommodate large peptide substrates upon binding [66]. 

The X-ray crystallography analysis of the protein structure determined that ERAP2 is 3.08 angstroms [66]. ERAP2 is an aminopeptidase localized in the endoplasmic reticulum for antigen processing [4]. Specifically, it is involved in antigenic peptide repertoire shaping [67] for the HLA system [4]. It finely processes intracellular peptides before loading them onto the HLA molecules. Thus, ERAP2 is a critical component in HLA processing [68]. 

ERAP1 has a similar structure to ERAP2, except for a few differences. Crystallization studies have identified two conformations of the ERAP1 protein structure, allowing the protein to expose its internal cavity where processive activity occurs. ERAP1 can use this internal cavity to trim larger peptides that ERAP2 cannot [69]. Other key differences in the peptide-binding internal cavity for antigen processing contribute to the unique ERAP2 function. The final difference was highlighted by one study that indicated that ERAP1 contains Gln 181, while ERAP2 contains Asp 198, resulting in differences in the ERAP2 S1 domain pocket. This caused changes in the ERAP2 peptide trimming efficiency [66]. Therefore, the absence or presence of ERAP1 and ERAP2 could change the peptide repertoire of trophoblast or tumor cells, increasing immune tolerance and allowing for increased proliferation and invasion. 

### 4.2. ERAP1 and ERAP2 Function

Immunodetection studies have determined that ERAP2 localizes with ERAP1 with endoplasmic reticulum markers [4], forming a heterodimer. However, the crystallization of the ERAP2 protein did produce a homodimer structure, making it possible for ERAP2 to homodimerize in the endoplasmic reticulum, in addition to forming a heterodimer with ERAP1 [66].

Antigenic peptide processive activities are shared by ERAP1 and ERAP2 when they are colocalized with specific, interdependent roles for each [66]. ERAP1 and ERAP2 have complementary functions when they select substrates according to the N terminus and internal sequences. ERAP1 preferentially cleaves hydrophobic residues and peptides with the hydrophobic C terminus [70]. After initial antigen processing by ERAP1, ERAP2 preferentially removes N-terminal amino acids from the epitope precursor due to the Asp^198^ residue [66], showing a strong preference for processing and hydrolyzing basic residues Arg and Lys [4]. 

A variation in the ERAP2 protein structure was identified in ERAP2 SNP rs2549782. This SNP resulted in a missense mutation resulting in a change from lysine (ERAP2K) to asparagine (ERAP2N). A crystal structure analysis comparison between ERAP2K and ERAP2N determined that the lysine (Lys) residue in ERAP2K assumes a distinct conformation compared to the asparagine residue in ERAP2. This residue difference produced different conformational interactions with Zinc-coordinating Glutamate (Glu)^393^, and residues Glu^337^ and Glu^200^, critical residues involved in stabilizing the N terminus of the substrate to ERAP2. The further computational analysis of ERAP2 identified highly unfavorable electrostatic interactions with Lys^392^ and the N terminus of a bound Lys ligand, suggesting that this may interfere with transition state stabilization, ultimately leading to a reduced catalytic efficiency for ERAP2K. Additional structural differences were identified between the two isotypes of ERAP2, specifically within the capping of the S1 pocket. The peptide difference affected the stability of the S1 bag, leading to changes in specificity with peptide bonding [71]. 

Further studies in ERAP2 colocalization with ERAP1 suggest ERAP2 functions as an accessory aminopeptidase to ERAP1 [66]. One study observing the trimming process of ERAP1 and ERAP2 for free and HLA B*0801-bound precursors noted that heterodimer formation enhanced the catalytic activity of ERAP1 while reducing the activity of ERAP2. Specifically, precursor trimming occurred mainly from ERAP1 being activated upon dimer formation with ERAP2. Active ERAP2 alone has poor trimming activity towards free and HLA B*0801-bound precursors [72]. However, RNAi experiments assessing the ERAP2 effect on cellular antigen processing determined that each enzyme could function independently. This result suggests that the ERAP1 and ERAP2 processive effects are additive [4].

## 5. ERAP1 and ERAP2 Correlation to Pregnancy and Cancer

ERAP, a critical enzyme expressed in trophoblastic cells, is responsible for peptide trimming, blood pressure regulation, immune recognition, and postnatal angiogenesis, all of which indicate why this protein’s expression was explored in pre-eclampsia (PE) [73]. ERAP2 deficiency in trophoblast cells may be beneficial for experiencing a successful pregnancy since it has been shown that regular patients have lower ERAP2 transcripts compared to patients with PE, and increased ERAP2 expression in PE patients was associated with clinical severity [74]. ERAP1 and the complexes formed with another endoplasmic reticulum protein 44 (ERp44) have a significantly higher detectable expression level in pre-eclamptic pregnancies than in normotensive ones [75]. An altered ERAP2 expression was found in the first trimester placentas of women who later developed PE [73]. An increased maternal ERAP2 expression in severe PE was associated with increased systolic blood pressure and antihypertensive medications needed to lower the blood pressure [74]. Specific populations also demonstrated an association with rs2549782 with PE, such as African American, Australian, and Norwegian people [76]. rs2549782 and rs17408150 single nucleotide polymorphisms (SNPs) have been detected in Australian and Norwegian populations and are associated with a disposition toward PE [77]. However, the Chilean population has not demonstrated the rs2549782 and rs2248374 LD found in the African American population, nor do they exhibit the LD between rs2549782 and rs2548538 reported by Yao et al. that is consistent with most people around the world. Yao et al. also hypothesize that the Chilean ERAP2 haplotype structure may allow the expression of the major T allele in rs2549782-encoding Asparagine (Asn). This contrasts with the function of the rs2549782 SNP, which substitutes Asn for Lysine (Lys) [76]. However, it was initially reported that the amino acid change resulting in N392K in SNP rs2549782 alters the antigen process, presenting implications for immune tolerance during pregnancy. The 165-fold increased antigen presentation of hydrophobic amino acids of 392N protein compared to that of 392K protein could significantly impact the antigen presentation in trophoblastic tissues where the increased production would make “foreign” paternal-encoded protein of the fetus more visible to the maternal immune system. As a result, this could activate an increased immune response from the mother against the fetal tissue [78]. Ultimately, this implies that in Chilean populations, the ERAP2 haplotype could impact peptide trimming and antigen presentation [76]. 

The expression of ERAP1 and ERAP2 has been associated with other pregnancy complications, such as miscarriages. The contrasting effects between the expression of ERAP1 and ERAP2 were observed in a study performed on women with repeated implantation failure (RIF) and fertile women. The secretion level of ERAP1 was significantly higher in the peripheral blood of fertile women who had given birth in the past compared to those participating in vitro fertilization (IVF) due to RIF. In contrast, ERAP2 was secreted less in fertile women than in RIF patients who became pregnant after IVF. Additionally, the RIF patients, who miscarried after IVF, secreted more ERAP2 than those who became pregnant by traditional means [79]. 

Studies in the transcriptional and post-transcriptional regulation of ERAP1 and ERAP2 have also been investigated in a cancer immunity setting. While it was documented that the expression of ERAP1 and ERAP2 is frequently altered in tumors compared to that in normal cells, only a few investigations have been conducted on how the altered expression of ERAP genes affects tumor growth and anti-tumor immune response. Defects in ERAP expression and function were detected in various solid and hematological tumors, which include melanoma, leukemia-lymphoma, and carcinomas. Table 1 describes the other cancers associated with ERAP and the impact ERAP expression has on the tumor. Several changes involving the ERAP protein occur during malignant transformation. Such changes include a low expression of ERAP1 and ERAP2 proteins in the tumors regardless of the histotype, the downregulation of one or both enzymes in breast, ovary, and lung carcinomas, the upregulation of both ERAP proteins in colon thyroid carcinomas, and ERAP1 and ERAP2 imbalances in all the tumor histotypes. The heterogeneous expression of ERAP1 and ERAP2 in tumors at the protein level matched the observed mRNA level [70]. A study observing ERAP1 and ERAP2 association in modulating the immune response in choriocarcinomas noted that ERAP2’s role in determining the peptide repertoire for HLA presentation is involved in modulating the NK and T cell immune responses [80]. An absence of ERAP2 expression in human trophoblast cells potentially supports choriocarcinoma development. 

Abnormal HLA peptides found in pregnancy and cancer are associated with ERAP2 expression. The ERAPs and HLA class I peptides play a role in the genetic risk for Hodgkin’s Lymphoma. Significant interactions between HLA-A11, a marker of Hodgkin’s susceptibility, and ERAP1 SNP rs27038 have been found. ERAP1 SNP rs26618 and HLA-Cw2 have similar interactions. In addition, it was found that risk alleles can alter the ERAP expression; for example, the A allele of rs27524 Hodgkin’s Lymphoma results in high ERAP1 and low ERAP2 levels. Other studies found that ERAP1 expression influences CD8+ T cell effectiveness in melanoma [81]. HLA-E also had significant ties to the immune response and ERAP2. Increased ERAP2 expression is associated with increased HLA-E in gynecological cancers, and cytotoxic lymphocyte infiltration is lowered with HLA-E expression [82]. In pregnancy, increased HLA-G in EVTs is associated with increased HLA-F and ERAP2 expression [83].

## 6. Discussion

Despite all the information available, there is still a significant lack of understanding of how trophoblast and cancer cells combine abnormal HLA presentation, the modulation of immune cells, and other pathways to survive and evade the immune system. An understanding of these pathways is vital in the treatment of pregnancy complications, where improper immune regulation causes a fetal “rejection” that may trigger pre-eclampsia (PE), pre-term births (PTB), unexplained stillbirths, recurrent miscarriages, intrauterine growth restriction (IUGR), and other disorders that are still under investigation [84]. In addition to pregnancy complications, understanding immune regulation and fetal rejection may lead to new developments in cancer treatments. However, further research needs to be conducted to understand these mechanisms better. 

In particular, the impacts of ERAP1 and ERAP2 on pregnancy and cancer should be explored further. ERAP1 and ERAP2 expression throughout the trophoblastic cells significantly impact the maternal immune system, as the enzymes’ responsibilities consist of HLA class I binding peptide trimming and blood pressure regulation [73]. Clinical observations have further supported the relationship between ERAP2 and pregnancy complications, as miscarriages, PE, and other pregnancy-related complications are accompanied by an up-regulation of ERAP2 [79,85]. 

In cancer and autoimmune disorders, ERAP1 and ERAP2 are emerging molecules and double-edged swords regarding immune responses. ERAP1 and ERAP2 trim peptides for antigen presentation on HLA class I molecules within the ER [65], which is crucial in tumor cell and immune system interactions. ERAP1 and ERAP2 are potential anti-cancer targets activating immune responses against malignant cancers by promoting T and NK cell-mediated cytotoxic responses [86]. One study noted that a higher expression of ERAP2 in the immunoreactive tumor microenvironment of Squamous-cell lung cancer (SqCLC) patients is correlated with high levels of immune markers and cells that include PD-L1, CD47, CD8+ TIL, CD68+ macrophages, and NK cells, which are all positive indicators for immunotherapy [87]. ERAP1 expression furthers the cancer progression, such as in cervical carcinoma. 

In contrast, an increased risk of tumor progression and lymph node metastases in Dutch populations is significantly associated with ERAP1-coding SNPs [88]. ERAP1 expression supports tumor progression, but differences in the ERAP1 association with cancer exist among different ethnic groups. In non-small cell lung carcinoma, four SNPs were significantly associated with Han Chinese people but not the Polish ethnic group [89]. These findings support that precise ERAP1 trimming is necessary for HLA class I molecules. HLA genes vary significantly across ethnic groups; thus, the appropriate ERAP1 allele is required to present the optimum size of antigenic peptides in the cells [67]. ERAP2 expression in cancer tends to be opposite to ERAP1, as ERAP2 deficiency is correlated with cancer growth through immune evasion. 

Renal carcinoma, colon adenocarcinoma, melanoma, and ovarian cancer show less ERAP2, suggesting that ERAP2 expression harms cancer progression. The ERAP2 SNP rs2549782 results in the isoform ERAP2N, which increases ERAP2’s affinity for hydrophobic peptides 165-fold, making the cells more susceptible to immune surveillance due to altered HLA and peptide presentations [90]. ERAP2N promotes cell death in trophoblast cells by downregulating the cell survival genes and upregulating the cell death genes [91]. ERAP2N can modulate immune recognition and potentially target cancer. Our prior research supports this; it demonstrates that ERAP2N elicits a strong NK and T cell immune response towards the ERAP2N-expressing JEG-3 gestational choriocarcinoma cell line [90]. The presentation of cancer-specific antigens and T-cell infiltration is necessary for an effective antitumor immune response. ERAP1 inhibition results in the greater activation of T and NK cells and a substantial change in the peptides present on the cancer cell surface to immune effector cells [92,93]. Thus, the role of these ERAP enzymes in cancer immune responses indicates the emerging interest in cancer immunotherapy [86].

A particular interest when examining ERAP enzymes and cancer immunotherapy is the inhibitors of ERAP enzymes. One study that linked the activity of ERAP2 to increased efficacy of immune checkpoint inhibitor cancer immunotherapy suggested that the pharmacological inhibition of ERAP2 could provide significant therapeutic implications. Using a potent ERAP2 inhibitor, DG011A, to treat MOLT-4 lymphoblastic leukemia cells for selective ERAP2 inhibition, the study noted that the treatment induced significant shifts in the immunopeptidome. Specifically, they identified 20% of the detected peptides as 9-mers that were optimal ligands for MHC class I alleles carried by MOLT-4 cells due to them having similar sequence motifs and a predicted affinity as optimal ligands. Ultimately, this study proved that ERAP2 inhibition could induce the presentation of many new peptides for MHC class I ligands and regulate the immunopeptidome [94]. Another potential mechanism is introduced when examining the increase in the NK and CD8+ T cell-mediated antitumor immune response with a reduced expression of ERAP1 in the tumors [95,96,97]. Peptide repertoire changes are present in immune effector cells, and the resulting increase in intra-tumoral NK and CD8+ T cells is caused by ERAP1 [95]. In addition, it has been shown that the NK cell-mediated killing of tumor cells can be induced through activity signals in the context of cross-reactivity between the peptides and NK cell-inhibitory receptors. In hepatocellular carcinoma cells (HCC), NK cells tolerate ERAP-deficient HCCs and the prompt killing of ERAP1-expressing HCCs. Therefore, the regulation of ERAP1 in activating the ligands on tumor cells and inhibitory receptors on NK cells could signal development for NK cell-based immunotherapy. An additional mechanism for treatment may be found in the neutralization of immune checkpoint inhibitors. Patients with the luminal subtype of bladder cancer showed an improved response to anti-PD-L1 immunotherapy in the setting of low ERAP2 expression [98]. A similar enhanced response of anti-PD-L1 was also seen in an ERAP1-deficient mouse transplantable tumor model [70,99].

Another future study path that could be elaborated on is the relationship between ERAP2 and autophagy. Autophagy is a process that occurs in pregnancy and cancer, which plays both a supportive role and a harmful role, although the reasoning behind this needs to be elucidated. In pregnancy, autophagy aids zygote development but inadvertently delays implantation, as cells can survive longer without nutrients [100]. Autophagy also helps the blastocysts to invade, regulate differentiation, and survive stressful conditions. On the other hand, high levels of autophagy are found in mothers with pre-eclampsia, hinting that autophagy is associated with some pregnancy complications [101]. In tumor environments, specifically leukemia, autophagy is detrimental and can be inhibited by specific proteins. However, autophagy inhibitors such as Matrine cause tumor regression in pancreatic cancers, which means autophagy benefits those cancers [57]. ERAP2 was associated with autophagy in pancreatic stellate cells (PSCs) within pancreatic ductal adenocarcinoma (PDAC). Elevated ERAP2 mRNA expression levels were found in PDAC tissues compared to those in normal pancreatic tissues, and an increased mRNA ERAP2 expression was associated with a poorer prognosis in PDAC patients. ERAP2 plays a role in activating PSCs, exacerbating tumor progression through autophagy. ERAP2 knockdown decreases the PSC’s ability to promote migration and the invasion of PDAC by inhibiting ER-derived autophagy. This autophagic response that activates the PSCs was driven by ER stress, while ERAP2 knockdown reduced ER stress and autophagy [102].

ER stress and its relationship with the ERAPs is another potential area of future study. ER stress was found to be somewhat beneficial during fetal development, but it becomes detrimental at higher levels. Mild ER stress is necessary for placental development. Still, excessive levels can lead to placental dysfunction in pre-eclamptic placentas and cause a reduced number of lysosomes in the trophoblasts [101]. ER stress advances the tumor progression in cancer environments and furthers the tumor’s development, growth, and adaptation in harsh environments [103]. ERAP1 and ERAP2 are likely involved in ER stress since they are found in the endoplasmic reticulum and are peptide-trimming enzymes. A lack of ERAP1 is associated with increased ER stress due to a build-up of untrimmed enzymes [104]. As described before, there are connections between ERAP2, autophagy, and ER stress in PDAC [102].

The research on the immune effects of pregnancy and cancer is incomplete, and more studies can be conducted on the complex effects of the conditions on the immune environment. A better understanding of the immune effects in pregnancy disorders and cancer can lead to new treatments for both conditions. As demonstrated here, ERAP1 and ERAP2 are associated with irregular peptides in the HLA system, several pregnancy complications, and cancers. Many SNPs in ERAP1 and ERAP2 are associated with pregnancy complications such as pre-eclampsia or tumor risks. Overall, research into the effects of ERAP1 and ERAP2 on pregnancy and cancer could yield a new understanding of the conditions and potential treatments for pregnancy complications and cancer, which affect many lives every day. 

## Figures and Tables

**Table 1 ijms-24-03454-t001:** Various cancers associated with ERAP, the bolded proteins, are broken down into the following bolded sections: disease type, its associated isoform, and its interaction regarding the aforementioned cancer.

Protein	Disease	Observation	Protein	Disease	Observation
**ERAP1**	murine T-cell lymphoma	Lymphoma rejected following inhibition of ERAP1 (Compagnone et al.) [70]	**ERAP1**	kidney renal clear cell carcinoma	ERAP1 found to be more expressed in this disease (Compagnone et al.) [70]
**ERAP1**	DAOY medulloblastoma	ERAP1 inhibition made disease more susceptible to NK cell-mediated killing (Compagnone et al.) [70]	**ERAP1**	cervical intraepithelial neoplasia	ERAP1 expression partially or totally lost with this disease (Compagnone et al.) [70]
**ERAP1**	alloreactive and non alloreactive lymphoblastoids	ERAP1 inhibition enhanced NK cell-mediated killing (Compagnone et al.) [70]	**ERAP1**	cervical squamous cell carcinoma	ERAP1 expression partially or totally lost with this disease (Compagnone et al.) [70]
**ERAP1**	murine colorectal carcinoma	Inhibition of ERAP1 led to tumor growth arrest and enhanced survival (Compagnone et al.) [70]	**ERAP1**	esophageal carcinoma	ERAP1 expression lost or reduced and associated with depth of tumor invasion (Compagnone et al.) [70]
**ERAP1**	cervical carcinoma	ERAP1 was expressed at high levels and associated with worse overall survival (Compagnone et al.) [70]	**ERAP1**	testicular germ cell carcinoma	ERAP1 less expressed in this disease (Compagnone et al.) [70]
**ERAP1**	non-small cell lung carcinoma	Several ERAP1 variants correlated with increased metastases and decreased survival (Compagnone et al.) [70]	**ERAP1**	uveal melanoma	ERAP1 less expressed in this disease (Compagnone et al.) [70]
**ERAP1**	acute myeloid leukemia	ERAP1 found to be more expressed in this disease (Compagnone et al.) [70]	**ERAP1**	adrenocortical carcinoma	ERAP1 less expressed in this disease (Compagnone et al.) [70]
**ERAP1**	stomach adenocarcinoma	ERAP1 found to be more expressed in this disease (Compagnone et al.) [70]	**ERAP1**	Kidney renal clear cell carcinoma	ERAP1 found to be more expressed in this disease (Compagnone et al.) [70]
			**ERAP1/** **ERAP2**	human lymphoblastoid cell lines	SNP rs75862629 of ERAP2 resulted in down-modulation of ERAP2 coupled with higher expression of ERAP1 (Paladini, 2018) [67]

## Data Availability

Not applicable.

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
