# Peer review of "A Comparative Review of Pregnancy and Cancer and Their Association with Endoplasmic Reticulum Aminopeptidase 1 and 2"

_ijms, 2023, doi:10.3390/ijms24043454_

Round 1

Reviewer 1 Report

The review is very substantial, but poorly organised. The various sections do not seem linked to each other, in particular the section on ERAP is not linked to the general part that precedes it. Some citations are missing (e.g. D'Amico et al doi: 10.3389/fimmu.2021.778103, to be added to ref 90). 

Instead of dividing the sections into 'similarities' and 'differences', the authors should discuss the various pathways and the aspects involved in them, discussing within them the similarities and differences between cancer and pregnancy. At that point, the discussion between ERAP and immunogenic tolerance would also be clearer. 

In the general part of ERAP, the notion that 25% of the population does not express ERAP2 should be better explained.

in Table 1, it is not clear what the authors mean by a positive effect of ERAP1

Do they mean that high expression is associated with good survival? 

Reviewer 2 Report

To authors,

The paper is well-written. This review is comprehensive and may provide important data for the future progress of this field. Only one point that I wish you to consider is: You compared the differences and similarities between cancer and trophoblast invasion/patho-etiology. You state that this strategy will be useful to detect/improve/enhance the understanding of many aspects of both cancer and pregnancy. I agree with this and as such I also performed research for 4-decades. This strategy or concept may not be new, let alone your own concept, I believe. If so, please state (add) the following meaning in a short phase, otherwise naïve readers might consider that this notion/concept (comparison the two and deduce some strategies) itself is your own/new/original finding.

Example, “many previous researchers recognized this concept and based on it, many new findings have been demonstrated: if substance X plays a crucial role in cancer invasion, then this might also play a role in trophoblast invasion, etc., vice versa. As such, better understanding of both pathophysiology has been made. Here we summarize,,,, “.  

Round 2

Reviewer 1 Report

the authors adequately responded and edited the text as requested

Author Response

Thank you for appraising our manuscript for publication in IJMS.

We thank the reviewers for their positive and constructive suggestions and have now dealt with each point raised and revised the manuscript accordingly. We believe the revised version of the manuscript is improved and thus hopefully will now be suitable for publication in IJMS. 

We hope that the revised version is now ready for publication and we look forward from hearing back from you.

Sincerely,

Dr. Eun Lee (Corresponding Author)

Virginia Commonwealth University